# RNA three-dimensional structure drives the sequence organization of potato spindle tuber viroid quasispecies

Jian Wu[1,2,3]*, Yuhong Zhang[1,2], Yuxin Nie[1,2], Fei Yan[1,2], Craig L. Zirbel[4], David M. Bisaro[3]

**1** State Key Laboratory for Managing Biotic and Chemical Threats to the Quality and Safety of Agroproducts, Institute of Plant Virology, Ningbo University, Ningbo, China, **2** Key Laboratory of Biotechnology in Plant Protection of MARA and Zhejiang Province, Institute of Plant Virology, Ningbo University, Ningbo, China, **3** Department of Molecular Genetics, Center for Applied Plant Sciences, Center for RNA Biology, and Infectious Diseases Institute, The Ohio State University, Columbus, Ohio, United States of America, **4** Department of Mathematics and Statistics, Bowling Green State University, Bowling Green, Ohio, United States of America

* wujian@nbu.edu.cn

**Data Availability Statement:** All processed data is presented in the paper or the supporting information. The raw deep sequencing data generated in this study has been deposited in the

## Abstract

RNA viruses and viroids exist and evolve as quasispecies due to error-prone replication. Quasispecies consist of a few dominant master sequences alongside numerous variants that contribute to genetic diversity. Upon environmental changes, certain variants within quasispecies have the potential to become the dominant sequences, leading to the emergence of novel infectious strains. However, the emergence of new infectious variants remains unpredictable. Using mutant pools prepared by saturation mutagenesis of selected stem and loop regions, our study of potato spindle tuber viroid (PSTVd) demonstrates that mutants forming local three-dimensional (3D) structures similar to the wild type (WT) are more likely to accumulate in PSTVd quasispecies. The selection mechanisms underlying this biased accumulation are likely associated with cell-to-cell movement and long-distance trafficking. Moreover, certain trafficking-defective PSTVd mutants can be spread by functional sister genomes in the quasispecies. Our study reveals that the RNA 3D structure of stems and loops constrains the evolution of viroid quasispecies. Mutants with a structure similar to WT have a higher likelihood of being maintained within the quasispecies and can potentially give rise to novel infectious variants. These findings emphasize the potential of targeting RNA 3D structure as a more robust approach to defend against viroid infections.

## Author summary

Our study investigates the world of RNA viruses and viroids, which are constantly evolving due to error-prone replication. These pathogens exist as quasispecies, featuring dominant master sequences and numerous variants. Understanding how new infectious strains emerge from this genetic diversity remains challenging. Focusing on potato spindle tuber viroid (PSTVd), targeted mutagenesis of specific stem and loop regions was performed to

NCBI Sequence Read Archive (SRA) under the accession number PRJNA1049893.The code is available at GitHub (https://github.com/tomwu1495/PSTVdQuasispecies).

**Funding:** This work was supported by a grant from the National Natural Science Foundation of China awarded to J.W. (32272483), a grant from the National Science Foundation to D.M.B. (IOS-1354606), and a grant from the National Institute of General Medical Sciences of the National Institutes of Health awarded to C.L.Z. (R01GM085328). The funders had no role in study design, data collection and analysis, decision to publish, or preparation of the manuscript.

**Competing interests:** The authors have declared that no competing interests exist.

generate mutant pools. Plants were infected with mutant pools, and progeny were analyzed following cell-to-cell movement and long-distance trafficking. Our results reveal that mutants with 3D structures similar to the wild type viroid are more likely to accumulate within quasispecies. This accumulation is associated with the ability to move between cells and travel long distances within the plant host. Even mutants with impaired mobility can be spread by functional counterparts within the quasispecies. This study highlights the role of RNA 3D structures in shaping viroid quasispecies evolution. Mutants resembling the wild type viroid have a greater chance of persisting within quasispecies and may give rise to new infectious variants. These findings offer insight into viroid quasispecies dynamics and identify RNA 3D structure as a promising avenue for resistance strategies.

## Introduction

Like RNA viruses, viroids exhibit high mutation rates during RNA replication [1–3], leading to the existence of diverse sequence populations known as quasispecies. Within a quasispecies, a small number of dominant master sequences coexist with a vast array of genetically diverse variants [4]. The master sequences and variants function as an entity, and the intricate population structure plays a crucial role in the adaptation and evolution of these pathogens, allowing them to rapidly respond to environmental challenges and host immune pressures [5]. Environmental changes can cause specific variants within viral quasispecies to become dominant, resulting in the emergence of new infectious strains [6]. However, accurately predicting the appearance of new viral pathogens and variants that can overcome resistance is currently not feasible.

The quasispecies concept highlights the dynamic nature of viral populations and their ability to explore sequence space through continuous mutation and selection. Quasispecies dynamics have been extensively studied in RNA viruses, such as human immunodeficiency virus (HIV), hepatitis C virus (HCV), and influenza virus, where they contribute to viral fitness, drug resistance, and immune evasion [7]. Viroids, which are small (250–400 nucleotides) circular RNA molecules, were previously believed to exclusively infect higher plants. However, recent discoveries have challenged this notion by detecting viroid-like RNAs in a broader range of hosts, including fungi [8–10]. These findings indicate the emergence of viroids as an important group of pathogens. Unlike viruses, they do not encode proteins and lack a protective coat. Viroids replicate in plant cells using host cellular machinery, including host DNA-dependent RNA polymerases, and spread between cells through plasmodesmata. They also travel systemically through the vascular system [11, 12]. Gaining insights into viroid quasispecies evolution is highly significant as it can aid in the development of novel approaches to combat viroid infections and advance our knowledge of RNA movement in the host organism.

While the existence and importance of quasispecies in RNA viruses and viroids is well established, the factors that shape sequence diversity within these populations are still not fully understood. It is clear that non-selective accumulation of variants alone cannot account for the observed patterns of quasispecies sequence diversity because certain variants are frequently observed while others are not [13–15]. Therefore, additional constraints and selective pressures must be at work. Existing theories involving fitness landscape, RNA secondary structure, and error threshold have limitations in addressing this issue. They overlook critical factors such as cellular tropism, transmissibility, and immune escape, and lack necessary mechanistic insights [16–19]. Novel concepts are needed.

RNA molecules can fold into complex three-dimensional (3D) structures, and these play critical roles in RNA function [20,21]. The relationship between RNA structure and function has been extensively studied, particularly in non-coding RNAs. Ribozymes, such as the group I introns and Ribonuclease P, as well as transfer RNAs (tRNA), have been particularly well-studied, highlighting the crucial role of specific structural motifs in their biological activities [22–25]. Viroids, such as PSTVd, are infectious, circular, non-coding RNA molecules. Their functionality relies on interactions with host factors through sequence and structural elements. Our research, using PSTVd as a model, has revealed the distinct and unidirectional nature of plasmodesmal gates that mediate viroid movement between different cell types. These gates interact with PSTVd RNA 3D motifs to govern its intercellular transport [26–33]. However, these previous studies focused on the relationship between local RNA 3D structures and the infectivity of individual PSTVd mutants [26–32], and the extent to which they influence the sequence structure of viroid quasispecies remains largely unexplored. Understanding the interplay between RNA structure and quasispecies sequence diversity could provide valuable insights into the constraints on both viroid and viral evolution. Moreover, it could have implications for our understanding of pathogenesis and the development of defensive strategies targeting RNA-based pathogens.

Studying natural quasispecies can yield limited information regarding the organization of sequences within a given sequence population, as the quasispecies has already undergone shaping and evolution. In this study, mutant pools were created by saturation mutagenesis of selected stem and loop regions in the secondary structure of PSTVd. Plants were infected with these mutant pools, and analysis of progeny after cell-to-cell movement and long-distance trafficking was carried out to explore the impact of RNA 3D structure on the sequence diversity of PSTVd quasispecies. Our findings revealed that, in stem regions, mutated purine/pyrimidine base pairs, which exhibit a structure more similar to WT purine/pyrimidine base pairs compared to purine/purine or pyrimidine/pyrimidine base pairs, account for only one-third of all possible base substitutions. However, after cell-to-cell movement and long-distance trafficking, purine/pyrimidine base pairs became the predominant mutated base pair type in PSTVd quasispecies derived from stem mutant pools. In loop regions, using JAR3D (Java-based Alignment of RNA using 3D structure) as a structural similarity scoring tool [34], we analyzed the compatibility of mutants within the PSTVd quasispecies derived from loop mutant pools with structural models predicted using the WT sequence. We observed that loop mutants whose sequence is consistent with the predicted WT loop 3D structure were preferentially accumulated in quasispecies, providing strong evidence supporting RNA 3D structure as a novel constraint on the sequence diversity of viroid quasispecies. The mechanisms underlying the biased accumulation of structurally correct mutants are likely associated with RNA movement, encompassing both cell-to-cell movement and long-distance trafficking. Our study suggests that variants sharing a structure similar to the WT are more likely to persist within quasispecies and can potentially lead to the emergence of new infectious variants. These findings underscore the potential of utilizing RNA 3D structure as a durable target for preventive interventions. They also have potential implications for the evolution of RNA virus quasispecies, given the emerging evidence linking RNA structure to viral infection [35–37].

## Results and discussion

### Purine/pyrimidine base pairs are the predominant mutated base pair type in PSTVd quasispecies derived from stem mutant pools

The secondary structure of PSTVd comprises 26 stems (S) and 27 loops (L). We selected one strand of three base-paired stems, namely S3, S15, and S26, for the preparation of saturated

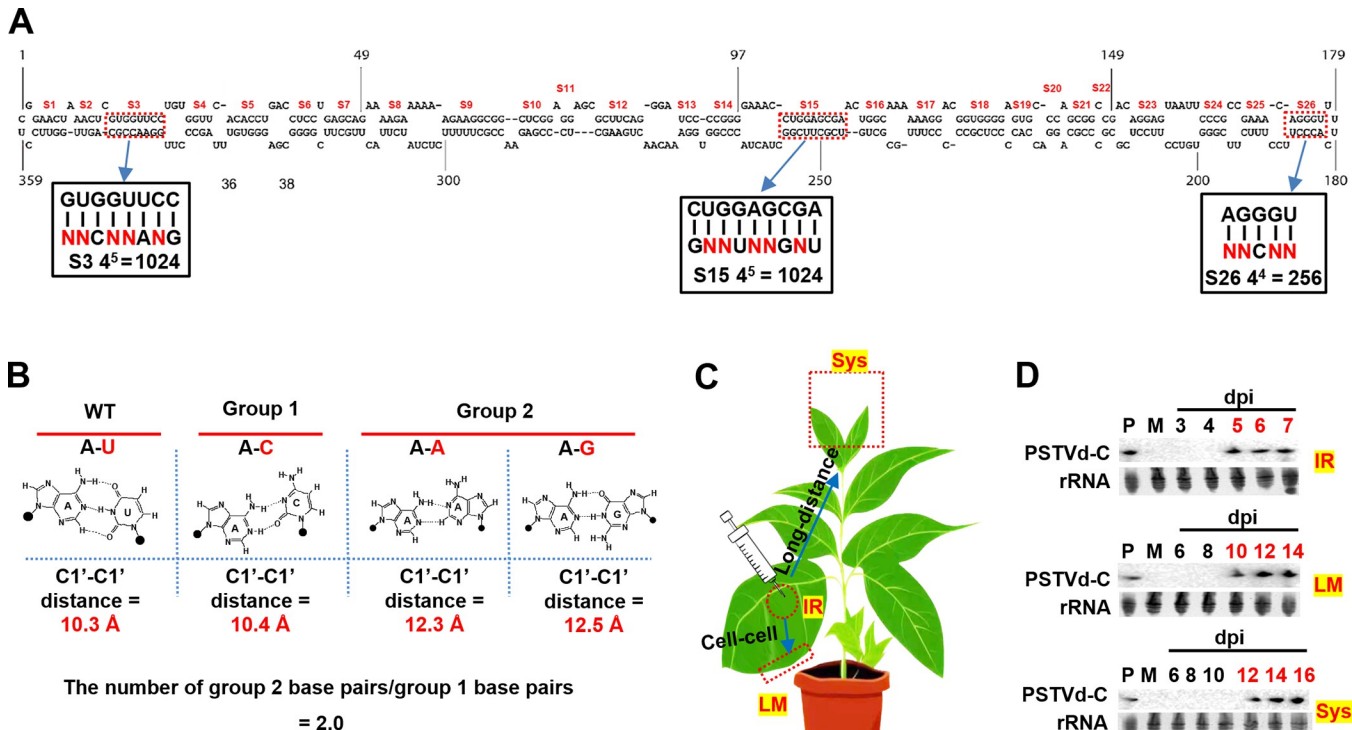

**Fig 1. Preparation of saturated mutant pools for selected stem regions, analysis of structural alterations caused by base substitutions in the stem region of the PSTVd secondary structure, and mutant pool inoculation strategy.** (A) The secondary structure of PSTVd and the design of saturated mutant pools on selected stem regions. The secondary structure of PSTVd consists of 26 stems (S) and 27 loops. The 26 stems are numbered. Three stems, S3, S15, and S26 were chosen for mutant pool preparation and are highlighted in outlined rectangles. The red Ns represent the degenerate regions, with N representing A, U, G, or C. The numbers of all possible sequences in the degenerate regions were calculated. Mutant pools were prepared through PCR using forward primers that contained these degenerate regions. (B) Principle of structural alterations caused by base substitutions in the stem region of PSTVd secondary structure. A base substitution in a canonical base pair within stem regions can lead to three possible mutated base pairs, as shown in this example. These include one purine/ pyrimidine base pair (group 1) and two other types of base pairs (purine/purine or pyrimidine/pyrimidine, group 2). Consequently, the ratio of group 2 base pairs to group 1 base pairs is 2.0. See text for the comparisons of 3D structure of different base pairs. (C) Inoculation strategy of stem mutant pools. Mutant pools of S3, S15, and S26 described in Fig 1A were delivered to the first two fully expanded true leaves of 3-week-old *N. benthamiana* plants through agroinoculation. Three biological replicates were included, with one plant per replicate. Using the S3 mutant pool as an example, samples were taken from the inoculated region (IR), the margin of the inoculated leaves (LM), and the upper systemic leaves (Sys) at various time points. (D) Analysis of the earliest infection time of S3 mutant pool at sites of IR, LM, and Sys. Samples collected from the IR, LM, and Sys sites at various time points were subjected to RNA blot analysis to determine the earliest time of infection. Circular form PSTVd (PSTVd-C), which is the functional form, was detected. Ethidium bromide staining of ribosomal RNA serves as a loading control.

mutant pools. These stems are highlighted within outlined rectangles (Fig 1A). Stems S3, S15, and S26 were chosen to represent the left, middle, and right regions of the PSTVd secondary structure, respectively. This selection aimed to preserve the critical pathogenicity and variable domains important for PSTVd infection and viroid-derived siRNA-mediated silencing of host genes [38]. Analysis of constraints on sequence diversity within these domains may be more complex, as nucleotide sequence (rather than structure) is critical for function. The degenerate regions are denoted by Ns in red color, where N represents A, U, G, or C. The total number of possible sequences within these degenerate regions was determined. It is worth noting that making alterations to the sequence of a large region can have a substantial impact on local structure, which can introduce complexity to base pair analysis. To mitigate this concern, mutations were exclusively performed on selected sites of a stem, with degenerate sites strategically interspersed by unmutated sites to minimize potential disruptions to the local structure (Fig 1A). A pNC-Green vector expressing full-length PSTVd intermediate strain RNA (from nucleotide 88 to 87) surrounded by two hammerhead self-cleaving ribozymes was used to

prepare mutant pools using degenerate primers [39]. Briefly, the experimental procedure involved conducting two separate PCR reactions (S1 Fig). The first reaction employed a degenerate forward primer encompassing the selected stem regions, in conjunction with a standard, non-degenerate reverse primer. The second reaction utilized a pair of non-degenerate primers. PCR products from the two reactions were mixed to allow homologous recombination, and resulting products were used to transform *E. coli* cells. To cover all possible mutants, a 10-fold excess of clones over the total number of possible degenerate sequences (10-fold clones) were collected and mixed. From these, mutant pools for one strand of each stem were obtained. To assess the quality of all mutant pools, the plasmid pools (inoculum) were utilized as templates for generating libraries through PCR amplification of the entire PSTVd genome. These libraries were subjected to deep sequencing on the Illumina NovaSeq 6000 platform. Stem sequences, comprising both the upper and lower strands in a 5' to 3' manner, were extracted and analyzed. All possible mutants were found to be present in the pools. It is important to mention that the number of unique sequences within each pool slightly exceeded the number of potential mutants. This can be attributed to errors introduced into unmutated sites during PCR and sequencing errors, which contribute to an increase in sequence diversity (S1 Table). But generally, the expected mutant sequences have significantly higher read numbers than unexpected ones. Unique sequences, the reads number for each sequence, and normalized read counts per thousand for each sequence are presented in S2 Table.

Before conducting experimental analysis on the PSTVd quasispecies derived from these pools, we considered the structural changes induced by different base substitutions within the stem regions of the secondary structure. Substitution of a single base within stem regions can result in three potential mutated base pairs. These include one purine/pyrimidine base pair (group 1) and two other types of base pairs (purine/purine or pyrimidine/pyrimidine, group 2). For instance, if a base substitution occurs on the U of an A/U base pair, it can generate A/C (purine/pyrimidine), A/A (purine/purine), or A/G (purine/purine) as three feasible mutated base pairs (Fig 1B). As a result, the ratio of group 2 base pairs to group 1 base pairs is 2.0. According to Leontis et al. [40], the 3D structure of an RNA base pair is determined by three factors: 1) the edges involved in base pairing, 2) the orientation of the base pairs, and 3) the C1'-C1' distance. RNA bases have three edges involved in base pairing, in either *cis* or *trans* orientations. In stem regions, bases primarily form base pairs through the Watson-Crick edge in a *cis* manner. Therefore, C1'-C1' distance is a key determinant of 3D structure variations in these base pairs. While a purine/pyrimidine base pair shares a similar C1'-C1' distance with other purine/pyrimidine pairs, they differ significantly from purine/purine and pyrimidine/pyrimidine pairs. For example, the C1'-C1' distance of an A/U base pair is 10.3 Å, closely resembling the 10.4 Å of an A/C base pair, but markedly different from the 12.3 Å of an A/A base pair and the 12.5 Å of an A/G base pair. Consequently, transitioning between group 1 base pairs results in subtle changes in the overall structure, whereas transitioning from a group 1 base pair to a group 2 base pair leads to significant structural alterations. Searches with the WebFR3D tool on the BGSU RNA site (http://rna.bgsu.edu/rna3dhub/) were conducted to identify helices with identical sequences and basepairs as S3, S15, and S26. An exact match for S26 was found (S2A Fig). However, no matches or single mutation matches were found for S3 and S15. This outcome is not surprising, given that these stems are longer than the typical RNA double helix. Constructing a model for these stems would simply reflect the expectation that they will closely resemble other RNA helices of the same size. In fact, analysis of resolved structures of helices revealed that a single base substitution typically brings about no significant alterations to the backbone structure. As an alternative approach, structural changes induced by replacing the UA cWW base pair, enclosed by two GC cWW base pairs, with GA or UU base combinations were presented. As anticipated, these two substitutions significantly

affect the replaced site while having minimal impact on the surrounding base pairs (S2B Fig). Nonetheless, even minor alterations to the backbone can carry significant biological relevance, as they have the potential to modify the helix's configuration and, consequently, its interactions with binding partners.

With an understanding of the relationship between different base pairs and their corresponding structures, we conducted experimental tests to investigate the involvement of RNA 3D structure in regulating the sequence organization of PSTVd quasispecies derived from the stem mutant pools. The mutant pools of S3, S15, and S26 were introduced into the fully expanded first two true leaves of three-week-old *Nicotiana benthamiana* plants via agroinoculation. In plant cells, PSTVd RNA is expressed from the 35S promoter within the pNC-Green vector, and unit length genomes are excised from the RNA by the flanking hammerhead ribozymes, initiating an infection. Three biological replicates were performed, with one plant per replicate. Samples were collected from the inoculated region (IR), the margins of inoculated leaves (LM), and the upper systemically infected leaves (Sys) at different time points (Fig 1C). These samples from the S3 mutant pool were used as an example to determine the earliest time point of infection using RNA blot analysis. The circular form of PSTVd (PSTVd-C), which is the functional form, was detected. Ethidium bromide staining of ribosomal RNA was employed as a loading control. The results indicated that the earliest infection occurred at 5, 10, and 12 days post-infiltration (dpi) for the IR, LM, and Sys sites, respectively (Fig 1D). Based on these findings, samples from these three time points were selected for library preparation for all other mutant pools.

RNA was extracted from IR, LM, and Sys samples derived from all stem mutant pools. Subsequently, libraries were generated through RT-PCR amplification of the complete genome. Three biological replicates were included for each sample from each stem mutant pool. In total, 30 libraries were prepared, which included the three inoculum pool libraries used to assess pool quality, mentioned above. It should be noted that IR samples may contain residual PSTVd DNA and positive-strand PSTVd RNA transcribed by the 35S promoter. To address these concerns, library preparation for the IR samples was conducted through RT-PCR amplification of the minus-strand using DNase-I digested RNA samples as template. In contrast, positive-strand PSTVd was detected in the LM and Sys samples. These libraries were sequenced on the Illumina NovaSeq 6000 platform. The deep sequencing data generated in this study has been deposited in the NCBI Sequence Read Archive (SRA) under the accession number PRJNA1049893.

The resulting deep sequencing data were processed using a Java-based pipeline. Unrelated sequences generated by incorrect PCR amplification were eliminated, resulting in more than 23,000 reads (ranging from 23,252 to 70,160) obtained for each library. It is noteworthy that the most prevalent variants observed within the sequenced PSTVd quasispecies are single or double mutants. These variants demonstrate minimal impact on the local structure, indicating the significant role of local structure in the accumulation of PSTVd mutants. The normalized Shannon entropy of the sequences from the three selected stems was calculated using the formula: $S_n = -\Sigma_i(f_i * \log_2(f_i)) / \log_2(N)$ [41,42]. Here, $S_n$ represents the normalized Shannon entropy, $f_i$ denotes the frequency of the ith sequence, and N represents the total number of reads in the dataset. The value of normalized Shannon entropy ranges from 0 to 1. High values indicate high sequence diversity, while low values indicate low sequence diversity. Compared to the inoculum pool, significantly decreased values of normalized Shannon entropy were detected in IR samples, which mimics a replication plus cell-to-cell movement system. Mutants detected in the LM samples are capable of moving from cell-to-cell, and a further decrease in normalized Shannon entropy was observed in LM samples across all three stem mutant pools. Mutants detected in the Sys samples have the ability to move systemically.

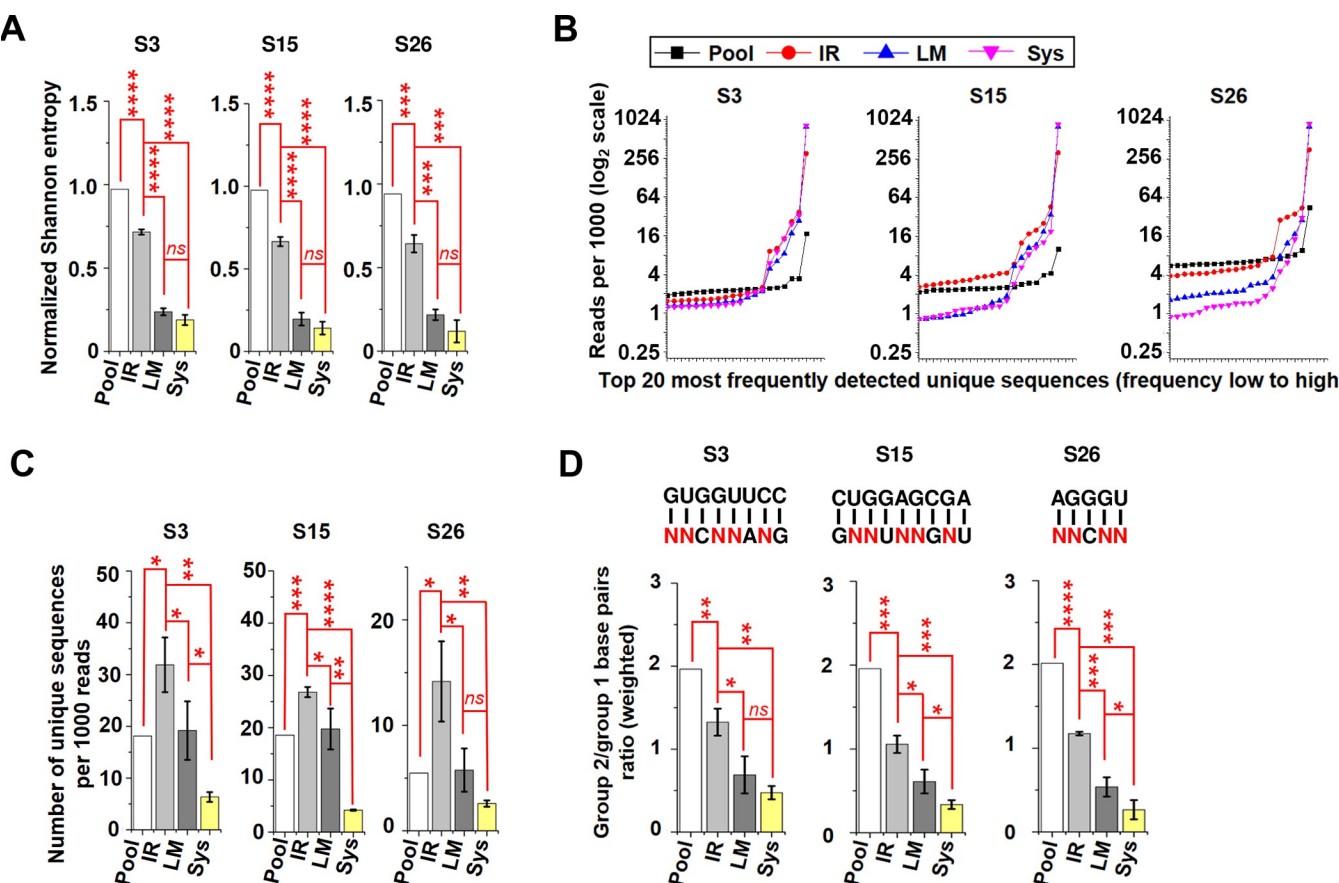

**Fig 2. Analysis of sequence diversity and types of base pairs in selected stem regions of PSTVd quasispecies derived from stem mutant pools.** (A) Analysis of the stem region sequence diversity of PSTVd quasispecies from mutant pool, IR, LM, and Sys samples. A total of 30 libraries were sequenced using the Illumina NovaSeq 6000 platform, and normalized Shannon entropy of the pool and the three selected loop sequences was calculated as described in the text. Values range from 0 to 1, with higher values indicating greater sequence diversity. (B) Read number of the top 20 most frequently detected variants from deep-sequenced libraries. The number of reads for the top 20 most frequently detected variants from the deep-sequenced libraries were normalized to a per-thousand basis and plotted on a $\log_2$ scale. (C) The number of unique sequences in deep sequencing libraries. The number of unique sequences of all deep sequencing libraries were normalized to a per-thousand basis and presented. Data are expressed as mean ± standard deviations (normalized Shannon entropy and number of unique sequences per thousand reads) or average values (reads number of the top 20 most frequently detected variants per thousand reads) for IR, LM, and Sys samples and a single value for the inoculum pools. Student's t-test was applied for comparisons: *p<0.05, **p<0.01, ***p<0.001, ****p<0.0001, *ns*, not statistically significant. (D) Sequences of both strands of stems S3, S15, and S26 were isolated from the deep sequencing dataset, and the base pairs formed by two strands of the same stem were analyzed. By comparing them to the WT sequence as a reference, the mutated base pairs at each position were analyzed. The weighted number of group 1 and group 2 mutated base pairs was calculated and compared between different samples (see text for details). Data are expressed as the mean ± standard deviations for IR, LM, and Sys samples, and as a single value for the inoculum pools. Student's t-test was applied for data comparisons: *p<0.05, **p<0.01, ***p<0.001, ****p<0.0001, *ns*, not statistically significant.

Compared to LM samples, a decreasing trend of normalized Shannon entropy was observed in Sys samples, although differences did not reach statistical significance (Fig 2A and S1 Table). The normalized read counts per thousand were determined for the top 20 most frequently detected variants in the deep-sequenced libraries. For the inoculum pool, a single value was reported since only one library was sequenced. In contrast, the IR, LM, and Sys samples were represented by average values obtained from the three biological replicates. In all cases, the WT sequence was the most frequently observed. Interestingly, a gradual increase in the percentage of WT sequence was observed from the pool library to IR, then LM, and the highest percentage of WT was detected in the Sys samples, consistent with the changes in normalized Shannon entropy values (Fig 2B and S1 Table). Furthermore, the number of unique sequences in all deep sequencing libraries was calculated and normalized to a per-thousand basis. It

should be mentioned that certain libraries, specifically the IR sample libraries, displayed a higher count of unique sequences than the expected number of mutants in the inoculum pool. This discrepancy can be attributed to errors introduced during PCR and sequencing, as well as mutations occurring at untargeted sites. In line with the changes in normalized Shannon entropy and the percentage of WT sequence in the quasispecies, a decrease in the number of unique sequences was observed from the IR library to the LM samples. The Sys samples exhibited the lowest number of unique sequences (Fig 2C and S1 Table). Therefore, cell-to-cell movement and long-distance trafficking can exert selective pressure on PSTVd quasispecies derived from the stem mutant pools.

A Java program was used to identify the base pair at each position for each unique sequence, and mutated base pairs were identified with the base pairs of the WT sequence as reference. Since the WT sequence, the most frequently detected sequence in all libraries, was used as a reference, it was not included in the following analyses. The weighted number of group 1 and group 2 base pairs was calculated using the formula: $WN = \Sigma(N_i * R_i)$, where WN represents the weighted number, $N_i$ represents the number of occurrences of group 1 or group 2 mutated base pairs in unique sequence i, and $R_i$ represents the total number of reads of unique sequence i. Using the weighted number, the ratio of group 2 base pairs to group 1 base pairs was determined and compared between different samples. As shown in Fig 1B, a single base substitution on a base of a purine/pyrimidine base pair results in two group 2 mutated base pairs and one group 1 mutated base pair. If mutations occur and accumulate randomly, the theoretical ratio of group 2 base pairs to group 1 base pairs should be 2.0. The ratios observed in all three stem mutant inoculum pool libraries were close to 2.0, indicating agreement with the theoretical expectation. Notably, a significant decrease in this ratio was observed in the IR sample, and a further decrease in LM samples across all three stems. In S15 and S26, a significant additional decrease was observed in the Sys samples compared to the LM samples (Fig 2D and S3 Table). Specifically, for S26, this ratio decreased from around 2.0 in the starting mutant pool to around 0.25 in the Sys samples, representing an 8-fold decrease. Therefore, mutated base pairs with a structure similar to the WT base pairs are more likely to accumulate in PSTVd quasispecies derived from the stem mutant pools. The mechanisms that mediate the biased accumulation of mutants are likely related to movement, including cell-to-cell movement and long-distance trafficking.

The composition of mutated non-WT purine/pyrimidine pairs was analyzed. Reads of eight purine/pyrimidine pairs (AU, UA, GC, CG, AC, CA, GU, and UG) in all IR, LM, Sys, and pool libraries were counted using the formula $RC = \Sigma(N_i * R_i)$, where RC represents the read count, $N_i$ represents the number of occurrences of each of the eight purine/pyrimidine pairs in unique sequence i, and $R_i$ represents the total number of reads of unique sequence i. The percentage of RC for each of the eight base pairs in the total RC of all purine/pyrimidine pairs was calculated. Compared to the pool libraries, no significant changes in the percentage of each of the eight bases were observed in all IR, LM, and Sys samples of S3 (S3A Fig), S15 (S3B Fig) and S26 (S3C Fig) libraries.

Stems composed of canonical A/U and G/C base pairs, as well as G/U pairs, serve as the backbone for basic RNA secondary structures, which are critical for various RNA functions, such as the self-cleavage activity of hammerhead ribozymes [43], substrate recognition by tRNA [44], and the regulation of translation for specific mRNAs [45]. Besides the role involved in the maintenance of RNA secondary structure, stem regions can also directly participate in critical biological processes. For instance, mismatches in the stem region of pre-microRNAs can disrupt the transfer of cleaved miRNAs from Dicer to Argonaute, resulting in compromised strand selection [46]. The roles of stem regions in PSTVd infection have not been thoroughly analyzed. However, it is evident that the formation of a proper rod-like structure is

crucial for the functionality of PSTVd [47,48]. In this study, saturation mutagenesis of selected stem regions was performed to investigate the role of structure in the sequence organization of these stems in quasispecies. As mentioned above, to minimize the effects of mutations on the local structure, degenerate sites were strategically placed alongside unmutated sites (Fig 1A). Furthermore, it is worth mentioning that the most abundant variants within the sequenced PSTVd quasispecies are single or double mutants, which likely have a relatively small impact on structure. Consequently, the potential impact of local structure alterations is expected to be minimal.

After cell-to-cell movement and long-distance trafficking, we observed a significant alteration in the composition of base pair types within the stem regions of PSTVd quasispecies derived from the corresponding mutant pools. Mutated purine/pyrimidine base pairs, which exhibit a high structural resemblance to the WT purine/pyrimidine base pairs, become the predominant base pair type, despite comprising only one-third of the base pairs in the inoculum pools. This observation is consistent with our previous findings, which indicated that PSTVd mutants responsible for transitioning from G/U pairs to other purine/pyrimidine base pairs are more likely to accumulate following long-distance trafficking [30].

We hypothesize that factors involved in spread are most important in shaping quasispecies. Interestingly, the most significant decrease in the group 2/group 1 base pair ratio (weighted) from pool to Sys sample was observed in S26 (8-fold change), followed by S15 (6-fold change), and S3 (4-fold change). In light of our hypothesis, a possible explanation for the smaller decrease observed at the S3 stem is that the left terminal region of the PSTVd secondary structure, which includes S3, is primarily involved in PSTVd replication rather than trafficking [48]. Although the function of S15 has not been fully characterized, its proximity to L15, a loop predominantly involved in PSTVd replication [31], suggests its potential role in replication as well. Since variants that fail to replicate will likely be selected out following initial inoculation, the effects of replication on PSTVd quasispecies in both the LM and Sys samples are expected to be minimal. Thus, the decrease in the group 2/group 1 base pair ratio (weighted) after cell-to-cell movement and long-distance trafficking suggests the potential involvement of S3 and S15 in these two processes in addition to replication, although further experimental validation is required to confirm this. Previous studies have revealed that S26 is a component of the RY motif, which serves as the binding site for the viroid RNA binding protein (Virp1) [49]. This protein interacts with IMPORTIN ALPHA-4 to facilitate the nuclear import of PSTVd [50]. While the role of this stem in the cell-to-cell movement and systemic trafficking of PSTVd remains unclear, our data also suggests its potential involvement in these two processes.

Taken together, our findings suggest that PSTVd variants in stem regions with correct or nearly correct RNA 3D structure are more capable of cell-to-cell and long-distance trafficking. Studies on both viroids and viruses suggest that plasmodesmal gates between different cell types are unique and unidirectional [26–32,51]. So, while plasmodesmal gates connect different cell types, they may selectively permit the passage of only those variants with correct or nearly correct structures. The distinct structural and functional characteristics associated with long-distance trafficking may impose an even more rigorous selection pressure on variants. This is because phloem loading of viroids, which is the critical step of long-distance trafficking, requires the involvement of multiple viroid structural motifs [30,32]. Together, these selective processes can shape the sequence organization of PSTVd quasispecies.

## Loop mutants that adopt a structure similar to the WT are preferentially accumulated in PSTVd quasispecies derived from loop mutant pools

The secondary structure of PSTVd consists of 27 loops, and nearly all of these play crucial roles in PSTVd replication and/or trafficking [52]. Although the loop regions are thought to be

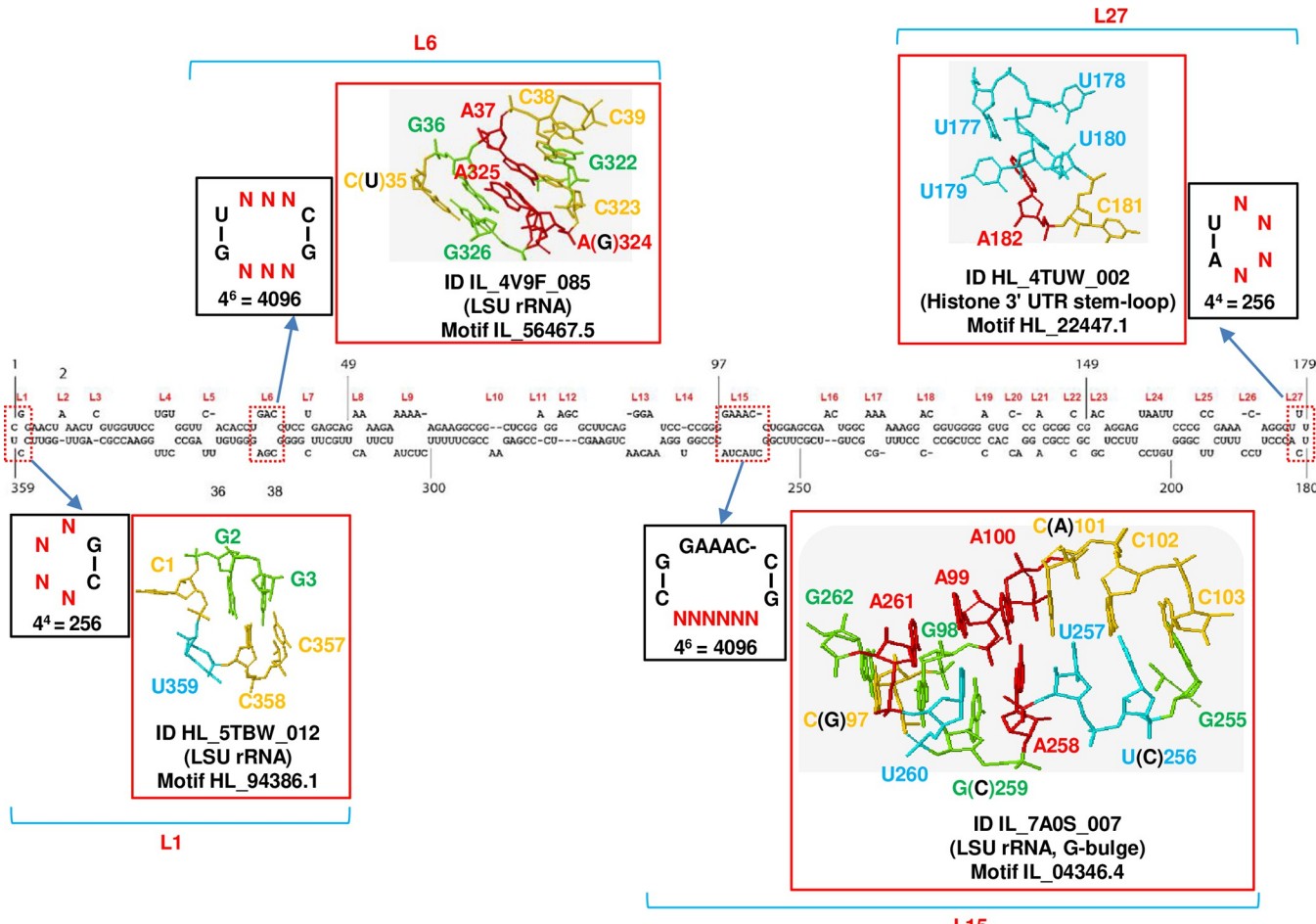

**Fig 3. Structural models of selected PSTVd loops and preparation of saturated mutant pools for those loops.** The PSTVd secondary structure consists of 27 loops (L), denoted by numbers in red (L1-L27). To analyze the constraints on sequence diversity within loop regions, saturated mutant pools were prepared for four selected loops: L1, L6, L15, and L27. Structural models of these four loops are outlined in red rectangles, along with the corresponding loop IDs from the BGSU RNA site (http://rna.bgsu.edu/rna3dhub/) and the names of the RNAs from which the models were derived. The structure models of L1, L6, and L15 originated from the large subunit ribosomal RNA (LSU rRNA). The structural model for loop 27 corresponds to the loop region found within the histone 3' untranslated region (UTR) stem-loop structure in animal cells. Bases in the structural models are labeled accordingly. In cases where model sequences differ from PSTVd sequences, the PSTVd sequences (in black) are provided in brackets. Saturated mutant pools were prepared for each of these loops, as outlined in black rectangles. However, due to technical limitations, saturating mutagenesis was only performed on the lower strand of L15. Ns represent nucleotides A, U, G, or C. The number of mutants in each pool was calculated and presented.

composed of unpaired bases, emerging evidence has shown that bases within loop regions can form non-canonical base pair to maintain a stable structure [40,53]. Previous studies have characterized the structure of several loops and analyzed their roles in the replication and trafficking of PSTVd [27,29,31]. To examine the influence of RNA 3D structure on the arrangement of sequences within loop regions of PSTVd, saturated mutant pools were created for four specific loops: L1, L6, L15, and L27 (Fig 3). These loops were selected because previous studies have thoroughly analyzed their structures using multiple approaches, which led to the proposal of promising structural models [27,29,31]. L6 controls the movement of PSTVd from palisade to spongy mesophyll [27]. Functional analysis of PSTVd L15 has revealed its crucial role in replication [31]. Hairpin L27 is involved in the movement of PSTVd from epidermal cells to mesophyll. The structural models for L6 and L15 are known from integral parts of the large subunit ribosomal RNA (LSU rRNA). More specifically, the structural model for L6 exhibits

conservation in helix 89 (H89) of bacterial and archaeal 23S rRNAs [27]. PSTVd L15 adopts a structure similar to archaeal and eukaryotic 5S rRNA loop E, also known as sarcin/ricin or G-Please bulge [31]. PSTVd hairpin L27 likely forms a 3D structure resembling the loop of a conserved hairpin found in the 3' untranslated region (UTR) of histone mRNAs in animal cells [29]. The structure of PSTVd hairpin L1 has not been analyzed in previous studies. Here, JAR3D (using RNA 3D Motif Atlas version 3.48) was used to predict the structural model of this hairpin loop using the WT (5'-CCUCGG-3') as the query sequence [34]. JAR3D is an online program that utilizes a combination of Stochastic Context-Free Grammars (SCFG) and Markov Random Fields (MRF) for its operations. Atomic-resolution RNA 3D structures serve as the foundation for constructing SCFG/MRF models, incorporating fundamental principles such as the isostericity of non-Watson-Crick (WC) base pairing and other base interactions [40]. Nonhomologous RNAs or distinct regions within the same RNA molecule may exhibit similar 3D motifs, and the recurrent geometries of these motifs tend to be more conserved than the constituent sequences themselves. JAR3D conducts searches within an RNA structure database, generating a compilation of known 3D motifs that align with a given query sequence. The most promising structural model of this loop, known as HL_5TBW_012, is also present in the LSU rRNA. Notably, this model shares the same sequence with PSTVd L1. The structural models of these four loops are displayed in outlined red rectangles (Fig 3), showcasing the corresponding loop IDs from the BGSU RNA site (http://rna.bgsu.edu/rna3dhub/), the standard structure names of the RNAs from which the models were derived (in brackets), and the motif groups to which they belong. The bases of the structural models were labeled and numbered as in PSTVd. In cases where model sequences differ from WT PSTVd sequences, the PSTVd bases (in black) are provided within brackets for reference (Fig 3). Saturated mutant pools were prepared for these loops, as outlined in black rectangles. However, due to technical limitations, saturation mutagenesis was only performed on the lower strand of L15. The number of mutants in each pool was calculated and presented.

JAR3D was utilized to search for structural models of single mutants within L1, L6, L15, and L27. Structural models with identical sequences were obtained for three single mutants of L1: C358U, C1U, and G2A (S4 Fig). However, no structural models with the same sequence were found for mutants of other loops. These three single mutations were noted to significantly impact the overall structure of L1. Since JAR3D only targets X-ray RNA structures with a resolution higher than 3.5 Å, the BGSU RNA group database (http://rna.bgsu.edu/rna3dhub/) of loops was also queried to investigate structural models with identical sequences for single mutants of other loops, thereby extending the search to structures solved by NMR, by cryo-EM, or with resolutions worse than 3.5 Å. Models with the same sequences were discovered for four mutants of L27: U179A, U179G, C181U, and C181A (S5 Fig). Similar to observations from L1, these four mutations significantly impacted the structure of L27. In the case of L6, an exact match was discovered during revisions to the paper, which exactly matches the WT sequence of L6 (S6A Fig). The exact match superposes extremely well with the model presented in Fig 3 (S6B Fig). No exact matches were found to L15, and no single mutation matches were found to L6 or L15.

Mutagenesis, inoculation, and RNA isolation were carried out following the same procedure as for stem pools, and libraries were prepared. Three biological replicates were included for each of the four loop samples at each of three sample sites (IR, LM, and Sys), resulting in a total of 40 prepared libraries, including four inoculum pool libraries. Unrelated sequences generated by incorrect PCR amplification were eliminated, resulting in more than 30,000 reads (ranging from 30,014 to 76,485) for each library (S1 Table). Deep sequencing data were processed using the same methods employed for libraries derived from stem mutant pools. Processing involved the calculation of normalized Shannon entropy (Fig 4A and S1 Table), read

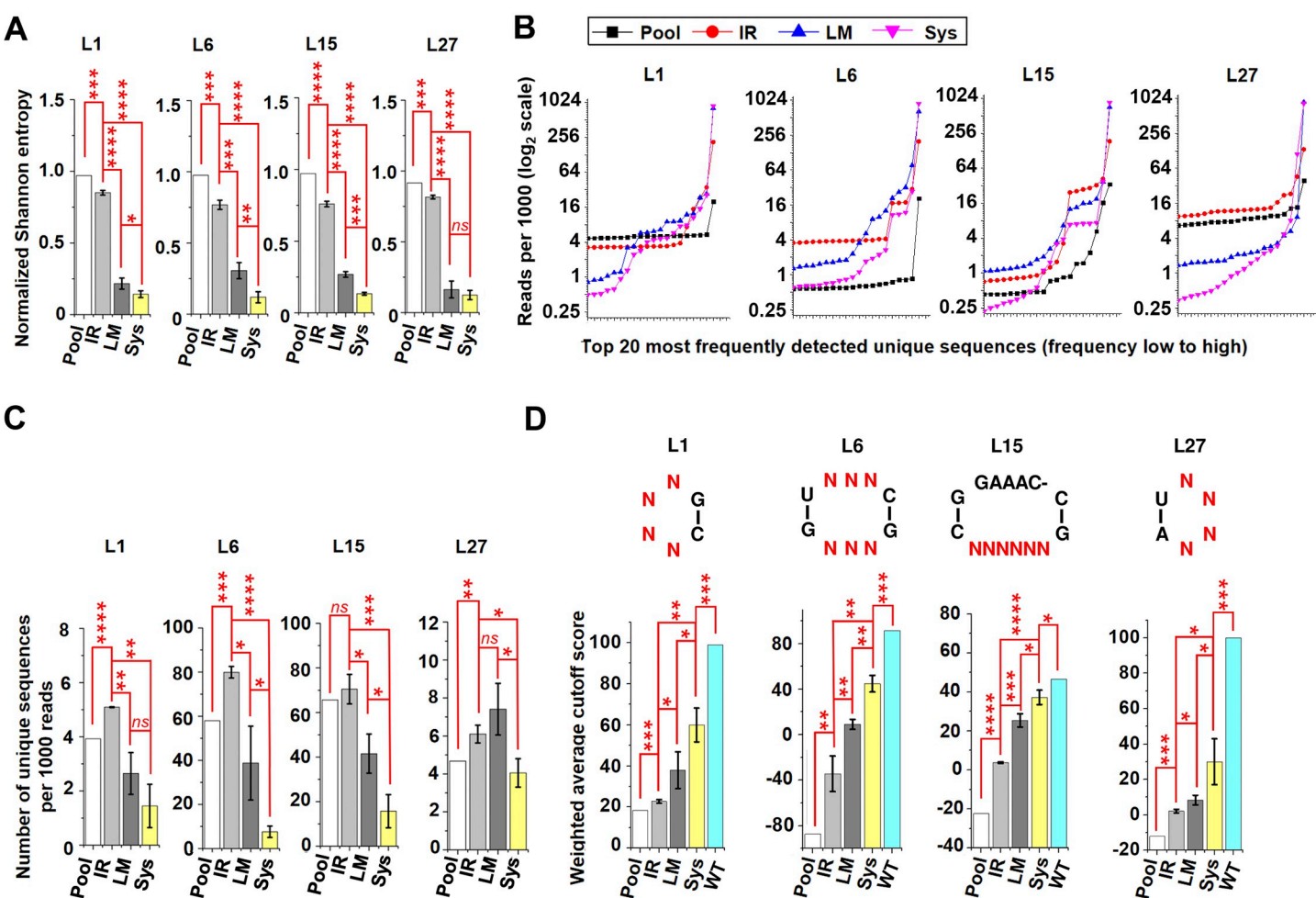

**Fig 4. Analysis of loop region sequence diversity in PSTVd quasispecies derived from loop mutant pools and structural modeling of variants** Using the same methods described in Fig 2, deep sequencing data from a total of 40 libraries derived from mutant pools of L1, L6, L15, and L27 were processed to calculate (A) normalized Shannon entropy, (B) read numbers per thousand reads of the top 20 most frequently detected unique sequences, and (C) number of unique sequences per thousand reads. (D) Using JAR3D the unique sequences of loop mutants in each library were aligned to their corresponding structural models. The output cutoff scores represent the compatibility of the unique sequences with the structural models. Higher cutoff scores indicate a higher level of compatibility. Weighted average cutoff scores of unique sequences in each library were calculated as described in the text. Data are expressed as the mean ± standard deviations for IR, LM, and Sys samples, and as a single value for the pools. Student's t-test was applied for data comparisons: *p<0.05, **p<0.01, ***p<0.001, ****p<0.0001, *ns*, not statistically significant.

numbers per thousand reads for the top 20 most frequently detected unique sequences (Fig 4B and S1 Table), and the number of unique sequences per thousand reads (Fig 4C and S1 Table). Unique sequences, the number of reads, normalized read counts per thousand, and cutoff score for each sequence are presented in S2 Table. As observed with stem libraries, the normalized Shannon entropy and the number of unique sequences decreased from inoculum pool to IR samples, followed by the LM samples, with the lowest values observed in the Sys samples. Furthermore, the percentage of WT sequence, which was the most frequently detected sequence in all libraries, increased from the pool to the IR samples, followed by the LM samples, with the highest percentage observed in the Sys samples. The only exception is that in L27, the percentage of the WT sequence in LM samples is slightly higher than in Sys samples. Consequently, the findings suggest that cell-to-cell movement and long-distance trafficking play a significant role in limiting the sequence diversity of PSTVd quasispecies derived from the four loop mutant pools.

The unique sequences of each loop in each library were aligned to their best-matching structural motif groups using the "Align sequences" function of JAR3D. As presented in Fig 3, the structural motif groups for L1, L6, L15, and L27 were HL_94386.1, IL_56467.5, IL_04346.4, and HL_22447.1, respectively. The WT sequence, which was the most frequently detected sequence in all libraries, was excluded from ~~the~~ subsequent analyses. Only unique mutant sequences were analyzed. JAR3D output cutoff scores served as a measure of how well a given sequence aligned with the predicted structure. A higher cutoff score in the JAR3D analysis indicates a greater level of compatibility between a specific sequence and the predicted 3D model structure. Weighted average cutoff scores were computed for unique sequences in each library using the formula: WC = $\Sigma$(Reads_i * CutoffScore_i) / $\Sigma$Reads_i. In this formula, WC represents the weighted average cutoff score, Reads_i represents the number of reads for unique sequence i, and CutoffScore_i represents the cutoff score for unique sequence i. Weighted average cutoff scores were employed to assess the overall compatibility of variants in the PSTVd quasispecies obtained from the loop mutant pools with the predicted structural models. As anticipated, the lowest weighted average cutoff scores were observed in the inoculum pool samples across all four loops. In comparison to the pool samples, there was a noticeable increase in the weighted average cutoff scores observed in the IR samples, and a further increase from the IR samples to the LM samples. The highest weighted average cutoff scores were observed in the Sys samples across all four loops, although these were still lower than the WT (Fig 4D and S4 Table). Because WT sequences are excluded from the analysis, the increases are solely due to increased compatibility between the mutant sequences and the structural model. Consequently, RNA 3D structure plays a crucial role in shaping the sequence organization of PSTVd quasispecies in loop regions. Mutant sequences that are more compatible with the structural model based on the WT sequence tend to accumulate preferentially. The mechanisms underlying the selection for these sequences are likely associated with cell-to-cell movement and long-distance trafficking.

In the quasispecies of RNA viruses, certain non-functional mutants can be carried by their functional sister genomes during both replication and trafficking processes [54]. Whether this phenomenon also applies to viroids remains unknown. In previous studies, the systemic movement ability of a considerable number of PSTVd L6, L15, and L27 mutants has been analyzed [27,29,31]. Consequently, we searched for these mutants in the Sys samples of our current study. Since only the lower strand of L15 was subjected to saturation mutagenesis, only mutants within this portion were included in this analysis. It was observed that certain trafficking-defective mutants were present in all three replicates of Sys samples across the three loops, although they generally accumulated to lower levels than trafficking-competent mutants (S5–S7 Tables). This suggests that trafficking-defective mutants in PSTVd quasispecies can also be carried by their sister genomes during systemic trafficking. Mutants found in all three Sys libraries were defined as carryable mutants, while others are considered non-carryable. In the case of loop 6 (S5 Table), among the 41 trafficking-defective mutants, 32 have a positive cutoff score, and nine have a negative score. All mutants with a negative score are non-carryable. Among the 32 mutants with a positive cutoff score, 23 are carryable and nine are non-carryable. For loop 15 (S6 Table), among the 18 mutants, 15 have a positive cutoff score, and three have a negative score. All mutants with a negative score are non-carryable. Among the 15 mutants with a positive cutoff score, 13 are carryable, and two are non-carryable. Fisher's exact test confirmed that in both the S6 and S15 libraries, trafficking-defective mutants with a positive cutoff score are more likely to be carried and reach Sys tissues ($p < 0.01$ and 0.05, respectively). No statistical analysis was performed on loop 27 since all mutants, including one with a negative cutoff score, are trafficking-competent (S7 Table). However, this mutant had the highest read count in the pool library and a relatively low read count in Sys libraries.

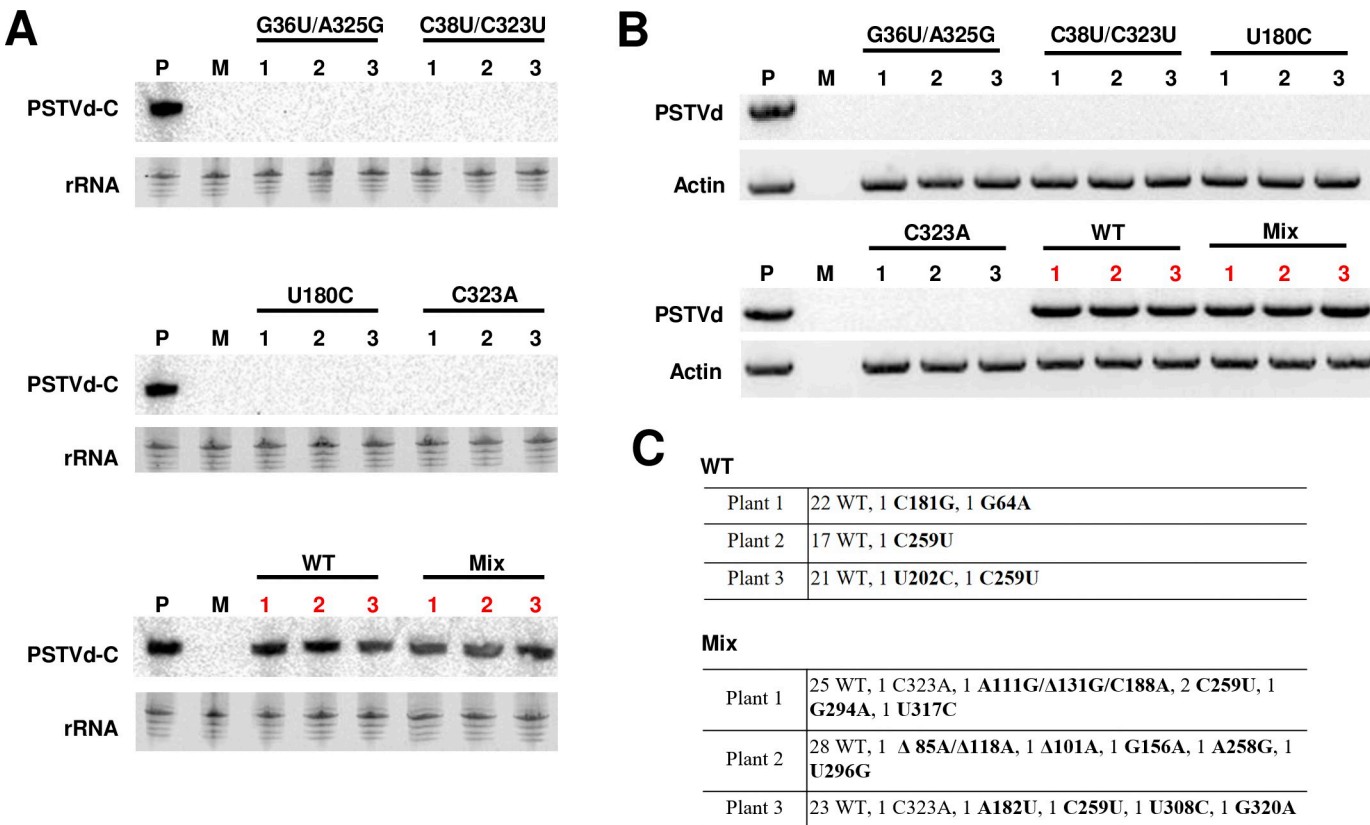

**Fig 5. WT PSTVd can carry a mutant with a nearly correct structure in systemic trafficking.** (A) RNA blot detection of PSTVd in *N. benthamiana* plants inoculated with WT and mutants. Two variants with positive cutoff scores (U180C and C323A), and two double mutants with negative scores (C38U/C323U and G36U/A325G) were used to test whether they can be carried by WT PSTVd in systemic infection. *In vitro* transcripts (300 ng each) of the four mutants and WT PSTVd were prepared and rub-inoculated onto the first two true leaves of two-week-old *N. benthamiana* plants, with three plants allocated for each treatment. Additionally, mixed co-inoculation (mix) was carried out by combining 37.5 ng of *in vitro* transcripts from each of the four mutants with 150 ng of WT *in vitro* transcripts (total 300 ng per plant). At 20 dpi, total RNA samples were prepared from systemic leaves and PSTVd was detected by RNA blot analysis, with ethidium bromide staining of rRNA serving as a loading control. Circular PSTVd (PSTVd-C), the functional form, is shown. A sample positive for PSTVd (P) and a mock (M) control were included. (B) RT-PCR, a more sensitive method, was also used to detect PSTVd with actin as an endogenous control. (C) Progeny from three plants of the WT and mixed co-inoculation groups were analyzed by Sanger sequencing. The number of sequenced clones is indicated, with novel mutations highlighted in bold.

To directly test whether certain trafficking-defective mutants can be carried by trafficking-competent sister genomes, two variants with a positive cutoff score, U180C (loop 27, cutoff score = 46.68) and C323A (loop 6, cutoff score = 51.82), and two double mutants with a negative cutoff score, G36U/A325G (loop 6, cutoff score = -29.40) and C38U/C323U (loop 6, cutoff score = -25.59) (S5 and S6 Tables), were used to co-infect *N. benthamiana* plants with WT PSTVd. These four mutants were shown to be trafficking-defective in previous studies [27,29]. Briefly, *in vitro* transcripts of these four mutants and WT PSTVd were prepared, and 300 ng of *in vitro* transcripts were separately rub-inoculated onto the first two true leaves of two-week-old *N. benthamiana* plants. Three plants were included for each mutant. Additionally, mixed co-inoculation was performed using 37.5 ng of *in vitro* transcripts from each of the four mutants and 150 ng of WT *in vitro* transcripts, totaling 300 ng for each plant. Total RNA was prepared from the top three fully expanded systemic leaves at 20 dpi, and RNA blot hybridization (Fig 5A) and RT-PCR (Fig 5B) were carried out to detect PSTVd. PSTVd was detected only in the WT and mixed co-inoculation groups, further confirming that the selected mutants are trafficking-defective. To test whether WT PSTVd can carry these mutants in systemic

trafficking, progeny in three plants of both WT and mixed groups were analyzed by Sanger sequencing. In the WT group, 24, 18, and 23 clones were sequenced for plants 1, 2, and 3, respectively. While novel mutations (in bold) were observed in all three plants, none of the four selected trafficking-defective mutants were detected. In the WT group, a total of 31, 33, and 28 clones were sequenced for plants 1, 2, and 3, respectively. Novel mutations (in bold) were observed in all three plants. C323A, but none of the other mutants, was detected once in plants 1 and 3 (Fig 5C). C323A has a relatively high cutoff score (51.82), suggesting that WT can indeed carry mutants with nearly correct structures in systemic trafficking.

Although L15 has been shown to participate in replication, our comprehensive analysis of progeny from the L15 mutant pool has revealed that most mutants in this loop were lost after cell-to-cell movement and long-distance trafficking. L6 and L27 play crucial roles in determining the movement of PSTVd from the palisade to the spongy mesophyll and from epidermal cells to palisade mesophyll, respectively [27,29]. Consistently, the movement of PSTVd from IR to LM samples, which requires these two types of cell-to-cell movement, led to the exclusion of mutants with a structure different from WT. In addition, a significantly increased weighted average cutoff score from LM samples to Sys samples was observed for both L6 and L27. This study also characterized the structure of the PSTVd L1, whose function has not previously been experimentally analyzed. After cell-to-cell movement and long-distance trafficking, there was a significant decrease in the sequence diversity of PSTVd quasispecies derived from the L1 mutant pool. Thus, our data suggest that all four loops may directly or indirectly affect the cell-to-cell movement and long distance trafficking of PSTVd. Indirect effects may include reduced replication and/or RNA stability. Furthermore, since samples from the IR, LM, and Sys were collected at different time points, a decreased replication rate or increased susceptibility to host defense mechanisms might lead to reduced sequence diversity in Sys samples.

Similar to the conclusions drawn from our analysis of stem mutant pools, plasmodesmal gates may selectively allow the transit of loop mutants with a similar structure to the WT, thereby shaping the sequence structure of PSTVd quasispecies. However, certain trafficking-defective loop mutants, mostly those capable of forming a structure similar to the WT, were detected in the Sys samples. This suggests that similar to the quasispecies of viruses, non-functional PSTVd mutants can be carried by functional sister genomes in systemic trafficking. However, a nearly correct structure may be a prerequisite for these non-functional mutants to be carried. The mechanisms that enable trafficking of defective viroid mutants are likely different from viruses, as viroids do not encode trans-acting proteins that can be utilized by other genomes within the quasispecies. One possible mechanism is that the functional genomes may utilize sequence and/or structural elements to open plasmodesmal gates, allowing defective mutants to move between cells. However, this hypothesis remains to be tested.

## Conclusions

Our study, for the first time, proposed and validated RNA 3D structure as a constraint on sequence diversity within a viroid quasispecies. Through interactions with host factors involved in cell-to-cell movement and long-distance trafficking, mutants that form a structure similar to the functional type (WT) may be allowed to cross cellular boundaries, resulting in reorganization of the sequence population. Since variants sharing a structure similar to the WT are more likely to persist and accumulate within the quasispecies, they serve as a likely source for the emergence of new infectious variants. These findings highlight the potential of targeting RNA 3D structure as an effective approach for achieving resistance to RNA-based pathogens.

## Materials and methods

### Preparation of mutant pools

Stem and loop mutant pools were generated by replacing the target sequence in the pNC-Green vector, which expresses plus-strand PSTVd intermediate strain, with two self-cleaving hammerhead ribozymes flanking the sequence. Briefly, to generate the mutant pools, two PCR reactions were conducted. One PCR utilized a degenerate forward primer, which covered the target region, and a normal reverse primer. The other PCR involved a pair of normal primers. The reverse primer used in the first PCR and the forward primer used in the second PCR were universal for all cases. However, the forward primer in the first PCR and the reverse primer in the second PCR were designed specifically to target the loops or stems of interest. The resulting PCR products were subjected to homologous recombination to prepare mutant pools. It is worth noting that, for L6, the upper and lower parts were mutated separately. The lower part was mutated through a PCR using the mutant plasmid pool prepared for the upper part as a template. In order to encompass all potential mutants, 10-fold clones were collected and combined. Primers were listed in S8 Table.

### Plant culture and agroinoculation

*N. benthamiana* plants were cultivated in a growth chamber at Ningbo University. Agroinoculation was utilized to deliver the mutant pools to the first two fully expanded true leaves of three-week-old *N. benthamiana* plants using 0.5 OD of agrobacterium. The experiments were conducted in triplicate, with each replicate consisting of one plant.

### Sampling and library preparation

Samples were collected from IR, LM, and Sys regions at 5, 10, and 12 dpi, respectively. The IR and LM samples consist of materials derived from two inoculated leaves, while the Sys samples comprise materials from the top two fully expanded leaves. Total RNA was extracted from *N. benthamiana* leaves using Trizol reagent (Invitrogen) and subjected to reverse transcription using the SuperScript Reverse Transcriptase IV kit (ThermoFisher Scientific). Subsequently, PCR amplification of the entire genome was carried out using Q5 High-Fidelity DNA Polymerase (NEB) to prepare libraries. It is important to note that the IR samples may contain residual PSTVd DNA and positive-strand PSTVd RNA transcribed by the 35S promoter. To address these concerns, library preparations for the IR samples were performed by conducting RT-PCR amplification of the minus-strand of PSTVd using DNase-I digested RNA samples as templates. In contrast, for the LM and Sys samples, positive-strand PSTVd was amplified.

### Deep sequencing and data analysis

The PCR product libraries were sequenced on the Illumina NovaSeq 6000 platform (Lianchuan Biotechnology Co. Ltd., China). The raw sequencing data underwent processing using a Java-based pipeline that involved the following steps: 1) A "SeqExtract function" was used to extract target stem and loop sequences. This function searches for a specific DNA sequence (8–10 bases just before the target sequence) within each line of the file and extracts a substring of the desired length starting from the position right after the identified DNA sequence. For loop sequences, the unmutated closing base pairs were also included. For stem sequences, both mutated and unmutated strands were extracted; 2) A "SequenceCounter" function was used to identify unique sequences of the target stem and loop sequences, and count the reads for each unique sequence. This function reads each line from the file. If a line starts with '>', it indicates the beginning of a new sequence; otherwise, the line is part of the current sequence. A

HashMap<String, Integer> was used to identify unique sequences; 3) A "QuasispeciesEntropy" was used to count total read number, the number of unique sequences, and calculate normalized Shannon entropy. This function iterates over unique sequences. For each unique sequence, the code calculates the frequency by dividing the count of occurrences of that sequence by the total number of reads. Normalized Shannon entropy was calculated using the formula $S_n = -\Sigma_i(f_i * log_2(f_i)) / log_2(N)$, where $S_n$ is the normalized Shannon entropy, $f_i$ is the frequency of the ith unique sequence, and $N$ is the total number of reads in the dataset; 4) A "BasePairAnalysis" function was utilized to analyze the base pairs formed by the unique sequences in the stem region. This function uses the WT sequence as a reference to exclude WT base pairs. The program identifies base pairs by comparing each pair in the input sequence with the corresponding pair in the reference sequence. It then counts how many times each base pair differs from the corresponding base pair in the reference sequence and outputs the results. Additionally, it counts the number of purine/pyrimidine base pairs and other types of base pairs for each sequence. The code is available at GitHub (https://github.com/tomwu1495/PSTVdQuasispecies).

The weighted number of group 1 and group 2 base pairs, as well as the composition of mutated non-WT purine/pyrimidine pairs, underwent analysis using Excel. The following two formulas were used: $WN = \Sigma(N_i * R_i)$, where WN represents the weighted number, $N_i$ represents the number of occurrences of group 1 or group 2 mutated base pairs in unique sequence i, and $R_i$ represents the total number of reads of unique sequence i; $RC = \Sigma(N_i * R_i)$, where RC represents the read count, $N_i$ represents the number of occurrences of each of the eight purine/pyrimidine pairs in unique sequence i, and $R_i$ represents the total number of reads of unique sequence i.

## PSTVd mutants and preparation of *in vitro* transcripts

The preparation of G36U/A325G, C38U/C323U, U180C, and C323A mutants involved utilizing a pRZ6-2 plasmid expressing PSTVd-I cDNA, as detailed in previous studies [27,29]. To prepare *in vitro* transcripts, the plasmid underwent linearization using Hind III (Cat # R3104, NEB) and served as a template for *in vitro* transcription with the T7 Megascript kit (Cat # AM1334, ThermoFisher Scientific). Subsequent steps included DNase I digestion to eliminate template DNA and RNA purification using the MEGAClear kit (Cat # AM1908, ThermoFisher Scientific).

## RT-PCR and progeny analysis

RNA samples were extracted and employed as templates for cDNA synthesis through reverse transcription (RT), utilizing the SuperScript Reverse Transcriptase IV kit (ThermoFisher Scientific). The PSTVd-R primer (5'-TGAAGCGCTCCTCCGAGCC-3') was applied for PSTVd, while the actin-R primer (5'-GCTCCTAGCGGTTTCAAGT-3') was used for actin. PCR reactions were executed with Phusion High-Fidelity DNA Polymerase (NEB). The primer sequences for PCRs were as follows: 5'-GGATCCCCGGGGAAACC-3' (forward) and PSTVd-R (reverse) for PSTVd, and 5'-ACCCTGTTCTCCTGACTG-3' (forward) and actin-R for actin. The resulting PCR products underwent 1.5% agarose gel electrophoresis. After purification, progeny sequencing was carried out using the TOPO TA Cloning Kit (Thermo Fisher Scientific).

## RNA blot

IR samples were collected at 3, 4, 5, 6, and 7 dpi. LM samples were collected at 6, 8, 10, 12, and 14 dpi, while Sys samples were collected at 6, 8, 10, 12, 14, and 16 dpi. To detect the infection

of selected mutants, samples were collected from the top three fully expanded leaves at 20 dpi. RNA samples were prepared using TRIzol Reagent (ThermoFisher Scientific). PSTVd infection was detected through RNA blot following the methods described by Zhong et al. [31].

## Supporting information

**S1 Table. Data summary of 70 deep sequencing libraries derived from three stem mutant pools and four loop mutant pools.** Samples were collected for deep sequencing at three sites, as illustrated in Fig 1C, namely IR (inoculated region), LM (margin of inoculated leaves), and Sys (systemic leaves). Three biological replicates (Reps) were included. Additionally, the pool sample used for the initial agroinoculation was also subjected to deep sequencing. After removing unrelated regions, at least 23,000 reads were obtained for each library. For each, the number of total reads, read number of wild type (WT), number of unique sequences, and sequence diversity (normalized Shannon entropy) are presented. IR and Sys samples are shown with yellow background.
(DOCX)

**S2 Table. List of unique sequences identified from each library.** Unique sequences identified from both stem and loop region libraries are listed (5'-3'). For stem region sequences, both upper and lower strands are presented. The number of reads, normalized read counts per thousand for each sequence, and cutoff scores for each sequence identified from the loop regions are also listed.
(XLSX)

**S3 Table. Analysis of mutated base pair types in PSTVd quasispecies derived from stem mutant pools.** Through the methods described in the legend of Fig 2D, mutated base pairs formed by two strands of the same stem in all three biological replicates (Reps) of each sample were analyzed. The weighted number of mutated group 1 or group 2 base pairs, and the ratio of the number of group 2 base pairs to the number of group 1 base pairs (weighted) were calculated and presented. IR and Sys samples are shown with yellow background.
(DOCX)

**S4 Table. Analysis of the weighted average cutoff scores of mutated loop sequences in the PSTVd quasispecies derived from mutant pools of selected loop regions.** Using the methods described in the legend of Fig 4D, the mutated loop sequences from all three biological replicates (Reps) of each sample were aligned to their corresponding structural models using JAR3D. The JAR3D output cutoff values were then utilized to calculate the weighted average cutoff scores. IR and Sys samples are shown with yellow background.
(DOCX)

**S5 Table. Analysis of the presence of previously functionally characterized loop 6 mutants in the Sys samples of the present study.** In a previous study, the trafficking ability of multiple L6 mutants was analyzed [27]. Their occurrence in Sys samples of the present study was determined. Trafficking-competent mutants are indicated in red, while the trafficking-defective mutants detected in the pool sample and all three Sys sample replicates (Sys-rep1, Sys-rep2, and Sys-rep3) are shown in yellow background.
(DOCX)

**S6 Table. The presence of previously functionally characterized loop 15 mutants in the Sys samples of the present study.** The trafficking ability of selected loop 15 mutants was previously analyzed [31]. Their occurrence in Sys samples of the present study was determined. Trafficking-competent mutants are indicated in red, while the trafficking-defective mutants

detected in the pool sample and all three Sys sample replicates (Sys-rep1, Sys-rep2, and Sys-rep3) are shown in yellow background.
(DOCX)

**S7 Table. The presence of previously functionally characterized loop 27 mutants in the Sys samples of the present study.** Our previous study analyzed the ability of selected loop 27 mutants in trafficking [29]. Their occurrence in Sys samples of the present study was determined. Trafficking-competent mutants are indicated in red, while the trafficking-defective mutants detected in the pool sample and all three Sys sample replicates (Sys-rep1, Sys-rep2, and Sys-rep3) are shown in yellow background.
(DOCX)

**S8 Table. Sequences of primers used in mutant pool preparation and library preparation.**
(DOCX)

**S1 Fig. Strategy for preparation of stem and loop mutant pools.** A pNC-Green vector, which expresses the plus-strand PSTVd intermediate strain (pNC-Green-PSTVd), was used as a template to create PSTVd mutant pools through PCR. To generate the mutant pools, two PCR reactions were performed. In PCR 1, a degenerate forward primer (Forward 1) covering the target region was used along with a regular reverse primer (Reverse 1). PCR 2 involved a pair of regular primers (Forward 2 and Reverse 2). Forward 2 and Reverse 1 were universal primers used in all PCRs. The resulting PCR products underwent homologous recombination, and the resulting products were transformed into competent *E. coli* cells. To cover all possible mutants, more than 10-fold clones were collected and combined, followed by plasmid preparation. The collected plasmids were then transformed into competent cells for Agrobacterium transformation. Finally, more than 10-fold clones were collected and mixed to prepare the mutant pools for agroinoculation.
(TIF)

**S2 Fig. Structural model of PSTVd stem 27 and illustration of structural changes induced by replacing UA cWW basepair enclosed by two GC cWW basepairs with GA or UU base combinations.** (A) The structural model of PSTVd stem 27 was obtained by querying the BGSU RNA site (http://rna.bgsu.edu/rna3dhub/). The model is sourced from PDB ID 1DQH. Positions 4 to 8 in chain A and positions 11 to 15 in chain B correspond to PSTVd positions 173 to 177 and 182 to 186, respectively. (B) The structural models of UA, GA, and UU enclosed by two GC cWW base pairs. These models were sourced from PDB ID 1DQH (positions 581 to 583 in chain A and positions 1261 to 1263 in chain B), PDB ID 4M4O (positions 12 to 14 in chain A and positions 46 to 48 in chain B), and PDB ID 4P43 (positions 28 to 30 in chain A and positions 19 to 21 in chain B), respectively. These models were overlapped to display the structural changes induced by replacing the UA cWW pair with GA or UU base combinations.
(TIF)

**S3 Fig. Composition of mutated non-WT purine/pyrimidine pairs on stem regions identified from corresponding stem libraries.** The reads for eight mutated purine/pyrimidine pairs (AU, UA, GC, CG, AC, CA, GU, and UG) in S3 (A), S15 (B), and S26 (C) libraries were calculated using the formula $RC = \Sigma(N\_i * R\_i)$. Here, RC denotes the read count, N_i signifies the occurrences of each purine/pyrimidine pair in unique sequence i, and R_i represents the total reads for unique sequence i. The percentage of RC for each of the eight base pairs in the total RC of all purine/pyrimidine pairs was then computed. Data for IR, LM, and Sys are presented as mean ± SD of three biological replicates, while data for pool libraries are expressed as a

single value.
(TIF)

**S4 Fig. Comparisons of structural models of wild-type (WT) L1 and its three single mutants.** Structural models for both the WT L1 and its mutants, C358U, C1U, and G2A, were generated using JAR3D. The sequences for the WT and mutants were provided above the models. Below the models, the corresponding loop IDs from the BGSU RNA site (http://rna.bgsu.edu/rna3dhub/) and the names of the RNAs from which the models were derived were presented.
(TIF)

**S5 Fig. Comparisons of structural models of wild-type (WT) L27 and its four single mutants.** A structural model for the WT L27 was constructed using JAR3D. Additionally, structural models for four mutants (U179A, U179G, C181U, and C181A) were generated by directly querying the BGSU RNA group database (http://rna.bgsu.edu/rna3dhub/). The sequences for both the WT and mutants were provided above their respective models. Below the models, the corresponding loop IDs from the BGSU RNA site and the names of the RNAs from which the models were derived were listed for the WT. For the mutants, only the loop IDs were presented.
(TIF)

**S6 Fig. A structural model with exact same sequence as L6 and its superposition with the structural model of L6 presented in Fig 3.** By directly querying the BGSU RNA group database (http://rna.bgsu.edu/rna3dhub/), a structural model (ID IL_7VPX_008) with the exact same sequence as L6 was obtained (A). This exact match superposes extremely well with the model of L6 presented in Fig 3 (B).
(TIF)

## Acknowledgments

This paper is dedicated to the memory of Dr. Biao Ding and Dr. Neocles B. Leontis, who imparted valuable advice at the very beginning of this study and ignited our inspiration to explore 3D motifs and their sequence variability in the viroid and other small RNAs.

## Disclaimer

The content is solely the responsibility of the authors and does not necessarily represent the official views of the National Institutes of Health.

## Author Contributions

**Conceptualization:** Jian Wu.

**Data curation:** Jian Wu, Yuhong Zhang, Yuxin Nie.

**Formal analysis:** Jian Wu.

**Funding acquisition:** Jian Wu, Craig L. Zirbel, David M. Bisaro.

**Investigation:** Jian Wu, Yuhong Zhang, Yuxin Nie, Craig L. Zirbel.

**Methodology:** Jian Wu, Fei Yan, Craig L. Zirbel, David M. Bisaro.

**Project administration:** Jian Wu, Fei Yan.

**Resources:** Jian Wu, Fei Yan.

**Software:** Jian Wu, Craig L. Zirbel.

**Supervision:** Jian Wu, Fei Yan.

**Validation:** Jian Wu, Yuhong Zhang, Yuxin Nie.

**Visualization:** Jian Wu.

**Writing – original draft:** Jian Wu.

**Writing – review & editing:** Jian Wu, Fei Yan, Craig L. Zirbel, David M. Bisaro.

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
