## [Decision Letter · Decision Letter 0]

23 Nov 2023

Dear Associate Professor Wu,

Thank you very much for submitting your manuscript "RNA three-dimensional structure drives the sequence organization of Potato spindle tuber viroid quasispecies" for consideration at PLOS Pathogens. As with all papers reviewed by the journal, your manuscript was reviewed by members of the editorial board and by several independent reviewers. In light of the reviews (below this email), we would like to invite the resubmission of a significantly-revised version that takes into account the reviewers' comments.

As reviewers point out some experimental evidence should be included to show directly whether  some of the mutants identified from NGS are movement defective and/or movement could be complemented by coinfection with wild type viroid. As reviewer #3 points out, all NGS raw data should be deposited into public repositories and the identifying deposited numbers should be included in the manuscript. Other comments from the reviewers could be addressed by revising figures and text.

We cannot make any decision about publication until we have seen the revised manuscript and your response to the reviewers' comments. Your revised manuscript is also likely to be sent to reviewers for further evaluation.

Sincerely,

Aiming Wang, Ph.D

Academic Editor

PLOS Pathogens

Savithramma Dinesh-Kumar

Section Editor

PLOS Pathogens

Kasturi Haldar

Editor-in-Chief

PLOS Pathogens

orcid.org/0000-0001-5065-158X

Michael Malim

Editor-in-Chief

PLOS Pathogens

orcid.org/0000-0002-7699-2064

Reviewer's Responses to Questions

**Part I - Summary**

Reviewer #1: In this manuscript, Wu et al. analyzed the changes in 3-D structural diversity of PSTVd mutant populations that occurred during the process in which the mutant pools with locally introduced random mutations spread from the infected cells to surrounding tissues and then to the upper leaves. As the result, the authors found that PSTVd mutant populations is subject to tertiary structure constraints during cell-to-cell movement and long-distance transport, and discussed that their quasispecies composition is shaped by these selective pressures.

The experiment is scientifically designed, and the progress of the experiment and the results obtained are explained in detail in a scientific manner.

The following points can be improved.

Page 9; “a gradual decrease in the number of unique sequences was observed from the pool library to the IR and LM samples”

Comment; In this experiment, at least the number of unique sequences was higher in IR than in pool library, so this statement may be misleading.

Page 16, L12-L17; “It was observed that certain trafficking-defective mutants (yellow shadow), although they generally accumulated to lower levels than trafficking-competent (red color) mutants, were present in all three replicates of Sys samples across the three loops (S4-S6 Tables), suggesting that trafficking-defective mutants in the PSTVd quasispecies can also be carried by their sister genomes during systemic trafficking”

Comment; Considering that most of the detected trafficking-defective mutants and trafficking-competent mutants differ by only one or two nucleotides, isn’t it possible that these trafficking-defective mutants were derived from trafficking-competent mutants by spontaneous mutations during replication in remote cells?

Legends for Figures, especially Fig 3, Fig 4, and Fig 5, are very long and seem to overlap with descriptions in the main text. It is preferable to avoid redundant explanations and simply state the necessary matters.

Reviewer #2: I approach this review as an RNA virus, but not viroid, specialist.

In this manuscript the authors perform saturation mutagenesis of 7 regions of the potato spindle tuber viroid and then use deep sequencing of input, inoculated region, margins of inoculated leaves, and systemically infected leaves to see which mutants are replication and/or movement competent. The authors also compare how similar the surviving mutants are to WT, at each stage, using Watson-Crick (in stems) and 3D RNA structure (in loops) metrics.

Overall this is a very nice manuscript. The techniques are elegant, I think that the data were analysed in a rigorous and thoughtful manner, and the manuscript is well-written and well-illustrated.

Reviewer #3: In this work Wu and colleagues used mutant pools of the viroid PSTVd to study the role of mutations on the systemic trafficking and infectivity of this sub-viral pathogen. Previous (similar) work from the late Professor Ding’s lab identified several motifs in the PSTVd genomic RNA that were basic for systemic trafficking and replication (10.1105/tpc.107.056606 and 10.3390/v10040160). Here the authors used a similar strategy, but instead they “challenged” a mutant pool and used high-throughput sequencing to understand the final outcome of the host-selected sequences in different types of tissues (which involve cell-to-cell and systemic trafficking). Overall the work is interesting, although it lacks experimental validation of the sequencing data and it could overall analyze the already produced data to a greater extent.

**Part II – Major Issues: Key Experiments Required for Acceptance**

Reviewer #1: Not applicable

Reviewer #2: o Basically every one of the 7 sets of mutated regions produces the same phenotype (a gradient of a reduction in the number of surviving sequences from Pool to IR to LM to Sys). The authors' interpretation is that every one of the mutated regions is involved in cell-to-cell and systemic movement. I wonder if this might be an over-interpretation, or perhaps too vague an interpretation to be useful (for example, it may be that every single nucleotide in the genome would show the same phenotype)? Is it possible that some of these movement phenotypes may be due to indirect effects (e.g. reduced stability of the molecule, reduced but not abolished replication, etc) rather than a specific direct role in movement? Related to this above point, there is also a time-course dimension to these samples as IR, LM and Sys were collected at different time points (5, 10 and 12 dpi), thus a reduced rate of replication or increased susceptibility to host defence, for example, might similarly lead to reduced representation in the Sys samples. Thoughts? No new experiments needed, but I suggest toning down the interpretation and/or discussing more what the interpretation really means.

o The proposal that some movement-defective mutants can be complemented by WT viroid is interesting but, as the authors discuss (p17), the mechanism underlying this is not obvious. It would be useful to at least more directly test this result (e.g. by coinfecting some of these mutants with and without WT viroid to check for complementation) to distinguish between a true complementation effect, as opposed to just the NGS approach being more sensitive to picking up some low level systemic movement of a movement-inhibited mutant.

Reviewer #3: Major Issues:

-Availability of raw data in public repositories. High-throughput sequencing data does not seem to be uploaded to a public repository, which is mandatory for publication in any journal. This is also key a service to the scientific community for the potential re-analysis of the author’s data.

-Overall the paper lacks a proper validation of the sequencing data. Neither of the PSTVd mutants identified was tested for its proper systemic trafficking abilities or maintenance of 3D structure.

-Another surprising lack of information in the manuscript is a further analysis of the sequencing results. The authors should consider including a map and analysis indicating the location of the identified mutations (including their presence within the already described domains in the PSTVd genome) that are important for the maintenance of both systemic trafficking and the 3D structure of the viroid circular form. This is key to visualize the location of mutations within the very characteristic PSTVd secondary structure.

-Overall quality of the figures. This is a somewhat minor issue, but the constant presence of compression artifacts in the figures and the lack of care in their preparation (boxes overlapping with images/text, PCR screenshots extremely pixelated even in the tiff version of the figures) led to me to include this comment here.

**Part III – Minor Issues: Editorial and Data Presentation Modifications**

Reviewer #1: Not applicable

Reviewer #2: o p2: "could be a more effective approach for antiviral strategies" -> "could be an effective approach for antiviral strategies"

o p9 line 7, text "IR samples, which mimics a replication plus cell-to-cell movement system" - is this the intended text, or are the IR samples supposed to measure replication _without_ a cell-to-cell movement component?

o p10 line 9: Should "mutated base pairs in unique i," read "mutated base pairs in unique sequence i,"?

o In Fig 1a in the inset at left, the "|"s are not aligned properly. Similarly for Fig 4.

o It would be useful to some curious readers to add a supplementary table showing the actual sequences of the top 20 unique sequences per region (Fig 3B, Fig 6B) along with their counts-per-1000. This would help interpretation of the results for people interested in the detail.

o Consider using a log y-axis scale for Fig 3B and Fig 6B, so that it would be possible to see the read counts for anything other than the WT viroid.

o In the stem mutants, are G:U pairs overrepresented among the non-WT purine/pyrimidine pairs that survive selection?

o Did the authors look for pseudorevertants - especially for the stem mutants, were there any cases where Watson-Crick base-pairings were restored by mutation on the opposite strand from the strand that was originally mutated? This could add extra weight to the 'maintaining WT structure' thesis.

o Regarding the text on p16 "except for one case in L27 (mutant 5'-UUAUAA-3'), all trafficking-defective mutants identified in all three biological replicates of the Sys samples had a positive JAR3D cutoff score": There are very few negative JAR3D scores in Tables S4-S6, so is the all-except-one absence of negative scores statistically significant or just random chance (Fisher's/chi-squared test)?

o On p12, I don't really understand what is meant by the text "our findings suggest that RNA 3D structure can _drive_the_sequence_organization_ of PSTVd stem regions primarily through cell-to-cell and long-distance trafficking" and "Together, these selective processes can lead to the _reorganization_ of sequences within PSTVd quasispecies." Consider rewording.

o On p12, the text on RNA virus movement is not really relevant to viroid movement since RNA plant viruses all encode movement proteins. Consider deleting or moving to Discussion?

o Some of the generalisations to RNA viruses (as opposed to viroids) are not supported and should probably be toned down e.g.

- p17 "Our study also provided novel insights into the constraints on sequence diversity within the quasispecies of viruses"

- p3 "Gaining insights into viroid quasispecies evolution is highly significant as it improves our understanding of viral evolution and quasispecies structure."

- p2 "These findings emphasize the potential of targeting RNA 3D structure as a more robust approach for antiviral strategies."

Reviewer #3: Minor issues:

-Lack of information in material and methods about the analysis of the sequencing data. The authors mentioned that “a Java-based pipeline” or a “A Java program” were used to analyze the data. The program in question is not described at all. This is an insufficient amount of information to replicate the authors results and should be explained in much further detail.

-The manuscript contains an unnecessary large number of figures. In my opinion Figures 1 and 2, 3 and 4, and 6 and 7 could be joined. That will leave the manuscript with a very appropriate number of 4 figures.

-In the analysis carried out in Figure 7 the cutoff score of the wt PSTVd sequence should be shown as a reference and to allow a proper comparison of the values.

-How could trafficking-defective mutants in the systemic tissues be carried out by functional sister genomes? Has any type of aggregation between viroid genomes been described in the literature? If not supported by the literature, I would guess that those sequences are able to move systemically due to similar interactions with transport proteins or a simple dilution of the mutant from the original infiltration/infection point.

PLOS authors have the option to publish the peer review history of their article (what does this mean?). If published, this will include your full peer review and any attached files.

Reviewer #1: No

Reviewer #2: No

Reviewer #3: No
---

## [Decision Letter · Decision Letter 1]

20 Feb 2024

Dear Associate Professor Wu,

Thank you very much for submitting your manuscript "RNA three-dimensional structure drives the sequence organization of potato spindle tuber viroid quasispecies" for consideration at PLOS Pathogens. As with all papers reviewed by the journal, your manuscript was reviewed by members of the editorial board and by several independent reviewers. The reviewers appreciated the attention to an important topic. Based on the reviews, we are likely to accept this manuscript for publication, providing that you modify the manuscript according to the review recommendations.

Sincerely,

Aiming Wang, Ph.D

Academic Editor

PLOS Pathogens

Savithramma Dinesh-Kumar

Section Editor

PLOS Pathogens

Michael Malim

Editor-in-Chief

PLOS Pathogens

orcid.org/0000-0002-7699-2064

Reviewer Comments (if any, and for reference):

Reviewer's Responses to Questions

**Part I - Summary**

Reviewer #1: The revised version has been appropriately improved according to the comments by reviewers.

In particular, the authors performed additional co-infection experiments using trafficking-deficient mutants and WT to substantiate their proposal that some of the trafficking-defective mutants can be carried to the upper leaves by WT. This finding is new in viroid, which made the manuscript more attractive.

Reviewer #2: The authors have carefully and diligently addressed all of my queries.

Reviewer #3: Not applicable

**Part II – Major Issues: Key Experiments Required for Acceptance**

Reviewer #1: None

Reviewer #2: (No Response)

Reviewer #3: In this revised version of their original manuscript Wu et al have addressed some of the issues of their first submission. Nevertheless, there are still some aspects that need to be addressed:

-The Java program used to analyze their sequencing data needs to be shared through Github (www.github.com) or any other repository for code.

-It is mentioned multiple times in the manuscript that some of the mutations studied affect PSTVd's 3D structure, yet, it is never shown how any of the mutations disrupt this. In the case of mutations in stems, the 3D structure of these stems is never shown. Could the authors please add to the manuscript: 1) the predicted 3D structure of the stems where mutations have been analyzed, and 2) the effect of some of the mutations on the predicted 3D structure for both stems and loops. At least some of the most dramatic alterations should be shown either in the main or supplementary figures.

**Part III – Minor Issues: Editorial and Data Presentation Modifications**

Reviewer #1: The following point can be further improved.

L475-L481; As mentioned above (Part I; Summary), the data shown in the new Fig. 5 supported strongly the notion that WT PSTVd can help systemic spreading of some of the trafficking-defective mutants with similar 3D-structure. However, if the authors add the data from the control plot, namely plants infected with WT alone, the results will be even stronger.

Minor point.

L445; trafficking-competent mutants, --- change comma to period.

Reviewer #2: (No Response)

Reviewer #3: (No Response)

PLOS authors have the option to publish the peer review history of their article (what does this mean?). If published, this will include your full peer review and any attached files.

Reviewer #1: No

Reviewer #2: No

Reviewer #3: No

Figure Files:

Data Requirements:

Reproducibility:

References:

---

## [Editor Report · Decision Letter 2]

22 Mar 2024

Dear Associate Professor Wu,

We are pleased to inform you that your manuscript 'RNA three-dimensional structure drives the sequence organization of potato spindle tuber viroid quasispecies' has been provisionally accepted for publication in PLOS Pathogens.

Best regards,

Aiming Wang, Ph.D

Academic Editor

PLOS Pathogens

Savithramma Dinesh-Kumar

Section Editor

PLOS Pathogens

Michael Malim

Editor-in-Chief

PLOS Pathogens

orcid.org/0000-0002-7699-2064
---

## [Editor Report · Acceptance letter]

28 Mar 2024

Dear Associate Professor Wu,

We are delighted to inform you that your manuscript, "RNA three-dimensional structure drives the sequence organization of potato spindle tuber viroid quasispecies," has been formally accepted for publication in PLOS Pathogens.

Best regards,

Michael Malim

Editor-in-Chief

PLOS Pathogens

orcid.org/0000-0002-7699-2064